# FERFusion: A Fast and Efficient Recursive Neural Network for Infrared and Visible Image Fusion

**DOI:** 10.3390/s24082466

**Published:** 2024-04-11

**Authors:** Kaixuan Yang, Wei Xiang, Zhenshuai Chen, Yunpeng Liu

**Affiliations:** 1Key Laboratory of Opto-Electronic Information Processing, Chinese Academy of Sciences, Shenyang 110016, China; yangkaixuan@sia.cn (K.Y.); xiangwei@sia.cn (W.X.); chenzhenshuai@sia.cn (Z.C.); 2Shenyang Institute of Automation, Chinese Academy of Sciences, Shenyang 110016, China; 3Institutes for Robotics and Intelligent Manufacturing, Chinese Academy of Sciences, Shenyang 110169, China; 4University of Chinese Academy of Sciences, Beijing 100049, China

**Keywords:** infrared-visible image fusion, transformer, deep learning

## Abstract

The rapid development of deep neural networks has attracted significant attention in the infrared and visible image fusion field. However, most existing fusion models have many parameters and consume high computational and spatial resources. This paper proposes a fast and efficient recursive fusion neural network model to solve this complex problem that few people have touched. Specifically, we designed an attention module combining a traditional fusion knowledge prior with channel attention to extract modal-specific features efficiently. We used a shared attention layer to perform the early fusion of modal-shared features. Adopting parallel dilated convolution layers further reduces the network’s parameter count. Our network is trained recursively, featuring minimal model parameters, and requires only a few training batches to achieve excellent fusion results. This significantly reduces the consumption of time, space, and computational resources during model training. We compared our method with nine SOTA methods on three public datasets, demonstrating our method’s efficient training feature and good fusion results.

## 1. Introduction

Due to technical limitations and the impact of the shooting environment, a single image captured by the same device often fails to provide a comprehensive description of the entire scene [1]. Therefore, image fusion techniques are emerging, in which infrared and visible image fusion is the most widely used. Infrared and visible image fusion (IVIF) produces fused images with complementary characteristics and richer information than either modality alone [2,3,4]. The resulting images are visually appealing and, more importantly, beneficial for practical applications such as medical diagnostics [5], remote sensing [6], automotive assistance [7], video surveillance [8], and wildfire monitoring [9].

Traditional infrared and visible image fusion methods are dedicated to finding the optimal representation of common features across modalities and designing appropriate weights for fusion [10]. For instance, multi-scale transform methods start by breaking down source images into features represented at multiple scales [11,12,13,14]. Similarly, sparse representation-based models begin by dividing source images into patches using the sliding window technique. Subsequently, these image patches are processed by sparse coding to derive sparse representation coefficients [15,16,17]. Low-rank representation (LRR) methods are also utilized to extract saliency features from source images, too [18,19]. After obtaining image features or representation coefficients, they will be reconstructed to produce the final fusion results according to delicately designed fusion rules. These fusion algorithms generally have the advantages of fast fusion speed and weak data dependence. However, manual feature selection and extraction are complex and cannot obtain good performance [20].

Deep learning has achieved tremendous success in various artificial intelligence applications in recent years, owing to its powerful non-linear fitting and feature extraction capabilities [3]. Researchers utilize public IVIF datasets or create [21,22], design, and build their own various deep neural network models for learning modal features and fusion strategies. After extensive training, these models can effectively extract information from infrared and visible images, generating information-rich fused images.

Unfortunately, many of the current advanced algorithms require a vast amount of space to store network parameters, as shown in Table 1, and their slow processing speeds make it difficult to serve downstream visual tasks. Compared to traditional approaches such as subspace analysis and sparse representation, deep learning-based methods have achieved notable improvements in fusion effects. However, their development has encountered bottlenecks: deep learning methods require the stacking of numerous convolutional modules to learn the common features of infrared and visible images, which have significant appearance differences; training these massive networks also necessitates a large number of strictly matched image pairs. Although there are public datasets like MSRS [23], M3FD [21], and LLVIP [24], they are still insufficient for models with millions or even tens of millions of parameters, such as RFN-Nest [25], CDDFuse [26], PIAFusion [23].

LRRNet [27] formulates the fusion task mathematically, and establishes a connection between its optimal solution and the network architecture that can implement it. It proposes a lightweight fusion network which avoids the time-consuming empirical network design by a trial-and-test strategy. However, the model has a large number of parameters, which limits further improvements in its speed. ReCoNet also explores a lightweight fusion network, inspired by traditional fusion knowledge priors [28]. It utilizes a simple max pooling channel and an average pooling channel, enabling the model to focus more on the textural features within the image. This approach is straightforward and effective, yet it might not be as flexible in feature extraction since the convolutional module in the model processes images treated with a simple attention mechanism rather than the source images. Moreover, due to their inherent characteristics, convolutional neural networks have a limited receptive field and lack the capability to perceive global features. This limitation prevents the establishment of long-range dependencies between features and leads to poor quality of the final fused image.

On the other hand, from the architectural perspective, visual transformers [29], with their unique self-attention mechanism that encodes global positional relationships, exhibit excellent global feature extraction capabilities. However, the computational cost of methods based on transformers is high, leaving room for further improvements considering the efficiency and performance trade-off in image fusion architectures. Therefore, we contemplate building upon ReCoNet by incorporating transformer modules [30] to enhance the model’s feature extraction capabilities and employing techniques like depthwise separable convolutions to reduce the model’s parameter count.

Overall, we employ two separate branches to extract features from infrared and visible images, respectively. Following [31,32], the exchange of a variety of information between different feature extraction channels can pre-fuse gradient and intensity information, serving as an enhancement of information before the next extraction. Hence, we have also introduced a parameter-sharing module between the two branches. To capture cross-modal, long-distance dependencies, we leverage transformer as our parameter-sharing module’s backbone. After obtaining the features, we use a convolutional module to further fuse and integrate information, ultimately outputting the fused image. Considering that different image features may have different scales, using traditional convolution layers with the same kernels cannot effectively represent source image information. In addition, to ensure the model’s lightweight nature, we employ three-channel dilated convolutions with different receptive fields for further processing of the features. The activation function for all layers is ReLU, except for the last layer, which uses Tanh [1].

To sum up, we proposes a lightweight recursive network, which can be trained on the public data set MSRS in 30 batches to fit, efficiently extract source image features, and generate fusion images with good visual effects. Specifically, we designed an attention module that uses traditional fusion knowledge prior guidance and channel attention to track the respective salience areas of the infrared and visible images to fit the network quickly. We also use an attention module with shared parameters to promote the early fusion of features; finally, we use a parallel dilated convolution module to learn features of different scales with different receptive fields. We iteratively train a set of parameters of this simple fusion module. This cyclic process reduces the number of network parameters and iteratively improves the image fusion quality.

## 2. Related Work

### 2.1. Infrared and Visible Image Fusion Based on Deep Learning

Due to the powerful nonlinear fitting capabilities, neural networks have been widely applied in infrared and visible image fusion, achieving performance far superior to traditional methods. Currently, the methods of infrared and visible image fusion based on deep learning can generally be divided into four types: CNN-based methods [33,34,35,36], GAN-based methods [37,38,39,40,41], AE-based [1,42,43,44,45] methods, and transformer-based [32,46,47,48,49] methods. CNN-based methods tend to focus on the design of loss functions, forcing the model to generate images that contain as much information from the source images as possible. GAN-based methods utilize an adversarial game between the generator and the discriminator to optimize the model. The generator produces images, and the discriminator judges whether the images are authentic or generated. When the discriminator cannot accurately judge, it is considered that the generated images meet the requirements. GAN-based models have a strong generation capability and can produce entirely new images, but their training process is complex and not robust enough [50,51,52]. AE-based methods force the decoder to generate images as close to the source images as possible during the training phase. When the two images are sufficiently close (optimal training loss), it is considered that the encoder can extract features from the source images well. In the testing phase, some specific fusion rules are applied to the results of the encoder for fusion and then decoded by the decoder to obtain the fused image. transformer-based methods have become popular in recent years, introducing attention mechanisms to encode the global positional relationships of images and overcoming the limitations of the convolution operation’s receptive field, but they also bring a high computational burden.

In recent years, some scholars have not limited themselves to the task of image fusion itself but have combined it with upstream and downstream tasks to guide image fusion or make it more robust and targeted. ReCoNet [28] introduces a geometric correction module to perform geometric compensation on the input pairs of infrared and visible images, thereby improving the model’s robustness. Similarly, RFNet [53] trains both multimodal image correction and fusion in a coarse-to-fine manner. SeAFusion [54] combines image fusion with downstream task of semantic segmentation, constructing a semantic loss that allows high-level semantic information to flow to the image fusion module, improving the effects of image fusion. IRFS [55] combines image fusion with saliency target detection, using the fused image generated by the fusion subnetwork as a third modality to drive precise prediction of the saliency map. Both enhance each other and are optimized together, achieving good experimental results.

### 2.2. Vision Transformer

The transformer was initially proposed by Vaswani and applied in the field of natural language processing [56]. It was first introduced to computer vision in 2020 [57]. IFT [58] was the first to introduce transformer into the IVIF field, proposing a multi-scale fusion strategy based on transformer to consider local and global information. Subsequently, PPT Fusion [59] improved upon the transformer by designing a patch transformer that converts images into a series of patches and then performs position encoding on each patch to extract local representations. It also designed a pyramid transformer to extract global information from the images.

YDTR [49] introduces a Y-shaped dynamic transformer architecture for infrared and visible image fusion. It designs a dynamic transformer module to extract local and global information simultaneously. SwinFusion [32] presents a universal image fusion method based on multi-modal long-range learning and the Swin transformer. It introduces self-attention-based intra-modal and inter-modal fusion units to capture global relationships within and between modalities effectively. CDDFuse [26] incorporates attention mechanisms into the image fusion framework. It uses CNN to extract modality-specific fine-grained features from infrared and visible images and employs the transformer’s long and short-term attention to extract modality-shared features, resulting in improved feature extraction and enhanced visual quality of the fused images. DATFuse [47] proposes an end-to-end dual transformer model to avoid manually designing complicated activity-level measurement and fusion strategies.

Until now, numerous transformer-based models have achieved excellent results in tasks such as image classification [60], object detection [61], segmentation [62], tracking [63,64], and multimodal learning [65]. For instance, in low-level computer vision tasks (e.g., denoising, super-resolution, and deraining), a pre-trained image processing transformer has outperformed CNN models [66]. Ref. [67] proposed a cross-scale mixing attention transformer-based model for hyperspectral and multispectral image fusion and classification. FD-Net [68] designed a feature distillation network for oral squamous cell carcinoma lymph node segmentation, which can effectively assist pathologists in disease screening and reduce workload. However, the application of transformers for mobile networks(with limited model size) significantly lags behind that for large networks. This is mainly because the computational overhead brought by most transformer models is not affordable for mobile networks.

Considering the heavy computational load of spatial self-attention, Wu et al. proposed a lightweight transformer architecture called LT for natural language processing tasks on mobile devices [69]. The model’s parameters are significantly reduced without sacrificing accuracy by incorporating long-short-range attention and a flattened feedforward network. Restormer [30] enhances the representation capabilities of transformers for high-resolution images by improving gated convolutional networks and multi-head attention modules. We drew inspiration from Restormer’s improvements on transformer architecture, adapting its concepts from image restoration to the task of image fusion.

## 3. The Proposed Method

### 3.1. Tri-Phase Attention Module

Texture features, such as edges, targets, and contours, play a crucial role in the fusion process. However, as the network deepens, texture features gradually degrade, leading to blurred targets and loss of details in the fused image. Existing work has focused on designing various attention mechanisms or increasing network width (such as dense and residual connections) to address this issue. In fact, some attention mechanisms have difficulty characterizing contextual features from the source images [28]; the increasingly large model architectures lead to a vast demand for computational resources and memory. We propose a tri-phase attention module that accelerates network convergence, suppresses information loss through an attention layer with traditional fusion knowledge priors, and enhances feature extraction capabilities through a channel attention layer.

The module includes a maximum pooling layer, an average pooling layer, a transformer attention layer, and a bias-free convolution layer. The maximum pooling and average pooling draw inspiration from traditional fusion strategies. The maximum and average values of each pixel position in the two images are calculated and then stacked together with the output of the transformer attention layer to serve as the input for the convolution layer.

Let A represent the tri-phase attention layer, and $I_{a}$ and $I_{b}$ represent the two input images, respectively. The following equation can represent this process: (1)$$A\left(I_{a},I_{b}\right)=\theta_{A}*\left[\max\left(I_{a},I_{b}\right),\mathrm{avg}\left(I_{a},I_{b}\right),\mathrm{trans}\left(I_{a},I_{b}\right)\right]$$
where ∗ represents the convolution operation, $\theta_{A}$ represents the parameters of the convolution layer, and [] represents the concatenation operation. As shown in Figure 1, the network calculates the attention maps $\sigma_{x},\sigma_{y}$ from the input image set {x,u,y} according to the following equation: (2)$$\sigma_{x}=A_{x}\left(x,u_{i}\right)\;\sigma_{y}=A_{y}\left(y,u_{i}\right)$$
where $A_{x}$ and $A_{y}$ represent the infrared and visible light attention layers, respectively, and $u_{i}$ denotes the fusion result obtained from the previous iteration.

Thanks to the tri-phase attention layer, our network can focus more on the features of each modality. Unlike traditional bi-phase attention layers (which only use max pooling and average pooling) [28], the tri-phase attention layer not only allows the network to converge quickly but also, with the introduction of channel attention, enables the network to more flexibly perceive the relationships between different channels, thereby enhancing the fusion effect [30].

### 3.2. Transformer Block

The structure of our transformer module is shown in Figure 2. It generally comprises two modules: a multi-head transposed attention module and a gated forward propagation module. Below, we will discuss the computational principles of these two modules in detail.

#### 3.2.1. Multi-Dconv Head Transposed Attention

In traditional transformers, the computational load primarily originates from the self-attention layer. The time and space complexity of the dot product between queries and keys grows quadratically with the input’s spatial resolution. For instance, for an image of size W×H, the complexity would be $O\left(W^{2}H^{2}\right)$, which is unacceptable for high-resolution images. However, the multi-head transposed attention module has a complexity that grows linearly, focusing on computing attention between channels rather than between spatial locations. This aligns perfectly with our image fusion task, which concentrates on the relationships between images of different modalities.

On the other hand, depthwise convolution further reduces the computational load and the number of parameters. Depthwise convolution performs convolution operations on each input data channel rather than convolutions across all channels.

For the input $X\in R^{H\times W\times C}$, after layer normalization, we obtain Y=LN(X). Then, through 1×1 convolution and 3×3 depthwise convolution, we compute our query (Q), key (K), and value (V) matrices: (3)$$\begin{array}{cc} \; & Q=W_{d}^{Q}W_{p}^{Q}Y \end{array}$$(4)$$\begin{array}{cc} \; & K=W_{d}^{K}W_{p}^{K}Y \end{array}$$(5)$$\begin{array}{cc} \; & V=W_{d}^{V}W_{p}^{V}Y \end{array}$$
where $W_{p}^{\left(\cdot \right)}$ and $W_{d}^{\left(\cdot \right)}$ represent the 1×1 convolution and the 3×3 depthwise convolution, respectively. Next, we reshape the query and key matrices so that their dot product produces a transposed attention map of size $R^{C\times C}$, rather than a traditional attention map of size $R^{HW\times HW}$. We first reshape the three matrices: (6)$$\begin{array}{cc} Q_{r} & =\mathrm{reshape}(Q) \end{array}$$(7)$$\begin{array}{cc} K_{r} & =\mathrm{reshape}(K) \end{array}$$(8)$$\begin{array}{cc} V_{r} & =\mathrm{reshape}(V) \end{array}$$

We obtain $Q_{r},K_{r},V_{r}\in R^{C\times HW}$, and then compute the attention map: (9)$$\mathrm{Attention}\left(Q_{r},K_{r},V_{r}\right)=\mathrm{Softmax}\left(Q_{r}\times K_{r}^{T}/\alpha \right)\times V_{r}$$
where the parameter α is a learnable scale parameter used to control the magnitude of $Q_{r}\times K_{r}^{T}$ before applying the softmax function. Thus, the output of the entire Multi-Dconv Head Transposed Attention (MDTA) module is obtained as follows: (10)$$\hat{X}=W_{p}\;\mathrm{Attention}\;\left(Q_{r},K_{r},V_{r}\right)+X$$

#### 3.2.2. Gated-Dconv Feedforward Network

For the input tensor $\hat{X}\in R^{C\times H\times W}$, the gated forward propagation layer can be described as: (11)$$\begin{array}{cc} Z & =W_{p}^{0}\;\mathrm{Gating}\;\left(\hat{X}\right)+\hat{X} \end{array}$$(12)$$\begin{array}{cc} \mathrm{Gating}(\hat{X}) & =\phi \left(W_{d}^{1}W_{p}^{1}\left(\mathrm{LN}\left(\hat{X}\right)\right)\right)⊙W_{d}^{2}W_{p}^{2}\left(\mathrm{LN}\left(\hat{X}\right)\right) \end{array}$$
where ⊙ represents the dot product, ϕ represents the GELU activation function, and LN stands for layer normalization.

### 3.3. Parameter Sharing Module

The structure of our parameter-sharing module is the same as that of the transformer module. Unlike the transformer attention branches, the parameters of the parameter-sharing module are shared between the infrared and visible light branches. This module can promote the early fusion of features, thereby improving the fusion effect. Specifically, for the *i*th iteration,
(13)$$\begin{array}{cc} \; & {\mathrm{base}}_{ir}={\mathrm{trans}}_{base}\left[x,u_{i}\right] \end{array}$$
(14)$$\begin{array}{cc} \; & {\mathrm{base}}_{vi}={\mathrm{trans}}_{base}\left[y,u_{i}\right] \end{array}$$
(15)$$\begin{array}{cc} \; & f_{in}^{i}=\left[x*\sigma_{x},u_{i},y*\sigma_{y},\;{\mathrm{base}}_{ir},\;{\mathrm{base}}_{vi}\right] \end{array}$$
where $x,y,u_{i}$ represent the infrared image, visible light image, and the fused image in the *i*th iteration, respectively; ${\mathrm{trans}}_{base}$ represents the parameter sharing module, [] represents the concatenation operation, * denotes the dot product, and $f_{in}^{i}$ indicates the input to the dilated convolution layer in the *i*th iteration.

### 3.4. Dilated Convolution Layer

We employ parallel dilated convolution layers to extract and fuse image features effectively. A set of dilated convolution layers with different dilation factors can expand the receptive field without losing neighborhood information. The three convolution layers across three channels all have the same 3×3 size convolution kernels, but they have varying receptive fields due to different dilation factors. As shown in Figure 1, the dilation rates from top to bottom are 1,2,3, respectively, with the receptive fields of the three parallel convolution channels being 3×3,5×5, and 7×7 respectively.

To express this more formally, we denote $f_{in}^{i}$ as the input to the dilated convolution layer in the *i*th iteration. The output feature map $f_{\mathrm{out}}^{i}$ of the recursive parallel dilated convolution layers can be updated step by step through the following formula: (16)$$\begin{array}{cc} \; & f_{\mathrm{out}\;}^{i}={\left\{C^{k}\left(f_{\mathrm{in}\;}^{i}\right)\right\}}_{k\in \{1,2,3\}} \end{array}$$(17)$$\begin{array}{cc} \; & C^{k}\left(f_{in}^{i}\right)=\theta_{C}^{k}*f_{\mathrm{in}\;}^{i}+b_{C}^{k} \end{array}$$
where, $\theta_{C}^{k}$ and $b_{C}^{k}$ respectively represent the weight and bias of the convolution layer with a dilation rate of *k*.

### 3.5. Recursive Learning

We adopted a recursive structure to replace traditional multilayer convolution to extract features from the source images from coarse to fine in a manner, as shown in Figure 3. The loop structure receives the fused result from the previous iteration as input for each cycle. Due to the lack of intermediate fusion results for the first iteration, we directly initialize the fused image as either the maximum or average value of the input infrared and visible light images. Compared to the linear stacking method, this approach can save 66% of parameters and mitigate the problem of information loss in deep networks. Due to the model’s minimal number of parameters and high processing speed, it is especially suitable for use on mobile devices.

### 3.6. Loss Functions

Our fusion loss function consists of three loss terms. The structural similarity loss $L_{\mathrm{SSIM}}$ is used to maintain the structure of the source images from the perspectives of illumination, contrast, and structural information. The intensity loss $L_{\mathrm{int}}$ is used to preserve the pixel brightness values of the source images. The gradient loss $L_{\mathrm{grad}}$ forces the fused image to keep consistent gradient information with the source images. Thus, our total loss function $L_{\mathrm{total}}$ can be expressed as: (18)$$L_{\mathrm{total}\;}=\alpha L_{\mathrm{SSIM}\;}+\beta L_{\mathrm{int}\;}+\left(1-\alpha -\beta \right)L_{\mathrm{grad}\;}$$
where, α and β are hyperparameters. Specifically, we hope that the fused image can have the same fundamental structure as the source images; hence, the structural similarity loss $L_{\mathrm{SSIM}}$ is defined as: (19)$$L_{\mathrm{SSIM}\;}=\left(1-\mathrm{SSIM}\left(u,x\right)\right)+\left(1-\mathrm{SSIM}\left(u,y\right)\right)$$

Similarly, the distribution of pixel intensity values in the fused image should achieve some balance between the infrared and visible light images; hence, the intensity loss $L_{\mathrm{int}}$ can be defined as follows: (20)$$L_{\mathrm{int}\;}={∥u-x∥}_{1}+{∥u-y∥}_{1}$$
where ${∥\cdot ∥}_{1}$ represents the $l_{1}$ norm. For gradient information, highlighted targets tend to have richer gradient information in the infrared or visible light images. For example, at night, infrared images highlight pedestrians, vehicles, and other targets, while other parts of the scene generally have lower intensity values; visible light images show more gradient information in well-lit areas. Therefore, if the fused image wants to retain the information from both source images, the gradient should equal the maximum gradient of the two source images. Hence, our gradient loss $L_{\mathrm{grad}}$ is designed as follows: (21)$$\begin{array}{c} L_{\mathrm{grad}\;}={∥{\mathrm{grad}}_{u}-\max\left({\mathrm{grad}}_{x},{\mathrm{grad}}_{y}\right)∥}_{1} \end{array}$$(22)$$\begin{array}{c} \mathrm{grad}\left(I\right)={∥\mathrm{sobelx}\left(I\right)∥}_{1}+{∥\mathrm{sobely}\left(I\right)∥}_{1} \end{array}$$
where sobelx and sobely are Sobel operators used to extract the gradient information of the image in two directions.

## 4. Experiments and Results

### 4.1. Dataset and Preprocessing

Dataset: To compare model performance, we selected three public datasets for our experiments: MSRS [23], TNO [70], and RoadScene [71]. Our training set is the MSRS (Multispectral Road Scene) dataset, which includes 1083 pairs of infrared and visible images. The test set consists of the MSRS dataset (20 pairs), the TNO dataset (20 pairs), and the RoadScene dataset (40 pairs). The TNO dataset contains grayscale images, while the MSRS and RoadScene datasets contain color images.Evaluation metrics: In this paper, we selected seven metrics as our evaluation criteria: information entropy (EN) [72], standard deviation (SD) [73], spatial frequency (SF), mutual information (MI), sum of the correlations of differences (SCD) [74], visual information fidelity (VIF) [75], and gradient-based fusion performance (Qabf) [76]. EN is a reference-free metric that measures the richness of information in the fused image from an information theory perspective. The higher the EN index, the richer the information contained in the fused image, which usually means the better the fusion effect. SD is also an independent indicator, usually used to evaluate the grayscale distribution and contrast of the fused image. As the variability in pixel intensities increases (which is reflected by an increase in SD), the image’s contrast also increases, making the visual content more distinguishable and engaging. SF denotes image detail clarity and spatial variation. The larger the SF, the richer the texture and edges. SF is an evaluation index based on image gradients. It evaluates the edge information in the fused image by comprehensively calculating the gradient in the horizontal spatial direction and the gradient in the vertical spatial direction of the fused image. MI determines the degree to which the fused image retains the source image information by calculating the sum of the dependencies between the fused image and the infrared image and visible image, respectively. The larger the MI value, the more information the fused image retains of the source image. SCD is a relatively nuanced metric designed to evaluate the quality of fused images by measuring the correlation of sum differences between them. Essentially, SCD assesses the spatial relationship and variance in intensity between pixels in different images, providing a measure of how changes in one image correspond to changes in another. This metric is particularly useful when comparing the information content and spatial detail of fused images to their source images. VIF is a metric that quantifies the quality of an image by assessing how accurately it preserves visual information from a reference image, based on the human visual system’s perception. Qabf is an evaluation index that calculates the saliency information from the source image retained in the fused image in each window from the perspectives of image contrast, clarity, and information entropy. The value of this indicator ranges from 0 to 1. The higher the values of the aforementioned metrics, the better the quality of the fused image. The comprehensive application of these metrics can thoroughly assess the quality of the fused images.RGB image processing: Our model can directly take color images as input for training and prediction without requiring manual code adjustments. Inspired by [28,34,77], we read color images in RGB format, then convert the images to the YCrCb space and extract the Y channel for fusion. After obtaining the fusion result, we use it as the new Y channel, combine it with the original Cr and Cb channels, and obtain the final color fused image. Our model can adaptively handle both color and grayscale images without the need for additional preprocessing operations.Training details: Parameters are updated using the Adam optimizer, with a learning rate of 0.001 and training for 30 epochs. The values of α and β are set to 0.5 and 0.1, respectively, and the model iterates three times. The training dataset is the MSRS dataset. Unless otherwise specified, all experiments are conducted on a computer with a single Nvidia RTX3090 GPU.

### 4.2. Qualitative Comparison

From Figure 4a, it can be seen that methods such as DenseFuse [78], PMGI [31], DIDFuse [79], SDNet [80], and U2Fusion [81] exhibit apparent ghosting phenomena around the edges of trees, whereas our method can avoid these flaws more effectively. The overall brightness of images produced by SDNet and U2Fusion is too high, while on the contrary, the overall brightness of images from PMGI, DIDFuse, and GANMcC [38] is too low (roads are not visible in Figure 4a). The LRRNet [27] method tends to reduce the contrast of targets, which is disadvantageous for target detection and tracking. Compared to ReCoNet, our method makes the pedestrians in Figure 4a,c and the stairs in (e) more clearly visible.

From Figure 5, it can be seen that the overall brightness of images from PMGI and SDNet is too high, while on the contrary, the overall brightness of images from DIDFuse and ReCoNet is too low. In MFEIF and GANMcC, the edges of pedestrian targets are blurred; in DIDFuse, U2Fusion, ReCoNet, and LRRNet, the contrast between infrared salient targets and the environment is lower. Meanwhile, our method can produce highlighted targets with precise contours and results with moderate brightness.

From Figure 6, it can be seen that the overall brightness of images from MFEIF [14], SDNet, U2Fusion, and ReCoNet is too high, while PMGI and GANMcC have an overall brightness that is too low and the targets are relatively blurry. In the images from DIDFuse and SDNet, the edges of the leaves are filled with a large amount of white ghosting, and there is considerable noise. Our method can produce fused images with moderate brightness which prominently display infrared targets and are relatively clear.

### 4.3. Quantitative Comparison

We selected seven evaluation metrics to comprehensively assess our method against nine other advanced fusion methods across three public datasets, focusing on the richness of information in the fused images (EN, MI, SF, Qabf), image contrast (SD), structural similarity to the original images (SCD), and visual effects (VIF), as shown in Table 2, Table 3 and Table 4.

From Table 2, we can observe that MFEIF performs well on metrics such as MI, VIF, and Qabf, indicating that its fused images are information-rich and visually appealing, as shown in Figure 4. DIDFuse performs well on EN, SD, SF, and SCD, suggesting its fused images are information-rich with higher contrast. Our method outperforms others on metrics like EN, SF, MI, VIF, and Qabf. It is second-best on the SD metric, indicating that the fused images produced by our method contain abundant information, have good visual effects, and possess high contrast, enabling the prominent display of infrared targets.

From Table 3, it can be seen that MFEIF and DIDFuse, which perform well on the TNO dataset, cannot effectively process images from the MSRS dataset. In contrast, DenseFuse is the best in terms of MI, VIF, and Qabf metrics, and second-best on the EN metric, indicating its superior ability to fuse information from color images. ReCoNet performs best on the SD metric, meaning its fused images have higher contrast, while our method excels in all seven metrics.

From Table 4, it is observed that DenseFuse performs well in metrics such as EN, SD, MI, VIF, and Qabf, indicating that its fused images contain rich information and have good visual effects. DIDFuse performs well on SF and SCD, suggesting its images contain rich texture details. However, as seen in Figure 6, its images are filled with a significant amount of noise at the edges of leaves and contain many white false edges in the sky in (d), leading to a potential misjudgment by the SF and SCD metrics regarding image quality. A similar issue occurs with GANMcC, where the EN metric might mistakenly consider noise as valid information. Our method performs best in SD, SF, MI, and VIF metrics, demonstrating that, compared to other methods, it produces images with less noise, more explicit images, moderate brightness, and good visual effects.

### 4.4. The Effectiveness of Recursive Training

We modified the architecture to a linear stacking approach to verify the recursive training methods’ effectiveness and compared it with our model. The training dataset is the MSRS dataset, and the test dataset consists of 20 pairs of images from the MSRS test set. See Table 5, Table 6 and Table 7.

In summary, the recursive training approach can significantly reduce the number of model parameters and is beneficial in preventing the loss of information due to overly deep models. However, there is no substantial improvement in GPU memory usage and runtime. This conclusion greatly differs from that of ReCoNet [28], which claims that a cyclical architecture reduces the time consumption by about 15%, the number of parameters by 33%, and GPU memory usage by 42%. For models with a depth of 3, the parameter count should be reduced by about 66%, as we have calculated.

### 4.5. Discussion of the Iteration in Attention Module

Figure 7 shows the impact of the attention module’s iterations on the fusion results. The attention map will be dot-multiplied with the image of the corresponding modality. After the concatenation operation, it will be used as the input of the dilated convolution module. It can be observed that with the progression of iterations, the infrared channel pays more attention to vehicles and pedestrians, while the visible light channel focuses more on bushes, the big tree on the right and in the distance. The information focused on by the two channels complements each other, extracting different features and laying the foundation for subsequent feature fusion to obtain a more information-rich fused image.

### 4.6. Ablation Studies

To demonstrate the effectiveness of the proposed module, we conducted ablation experiments on the MSRS dataset, with the results shown in Table 8. ‘ori’ refers to the model that lacks both the transformer attention branch and the shared feature fusion module, with the ReCoNet model serving as a reference; ‘+base’ denotes the model obtained by removing the transformer attention branch; ‘+trans’ represents the model with the shared feature fusion module removed. All models were trained on the MSRS dataset for 30 batches and underwent partial parameter tuning to demonstrate the model’s capabilities fully.

Adding either the transformer attention branch or the shared feature fusion module alone can significantly enhance the performance of the original model, with improvements across all seven metrics. When both modules are incorporated, there is a slight decrease in the model’s performance on SF and SCD. However, the fused images contain richer information (EN, MI increase), have greater contrast (SD increase), and offer improved visual effects (VIF increase).

### 4.7. Fusion Time Test

To better demonstrate the actual running speed of our model, we compared the fusion speed of 11 state-of-the-art fusion models using 20 pairs of images from the TNO dataset, with the results shown in Figure 8.

In the reproduction of U2Fusion, code was added to convert image precision from float32 to uint8, which slightly increased the fusion time for U2Fusion compared to the original. In the ReCoNet model, the m-Register registration module was activated instead of using the pre-trained model provided by the authors. For the replication of TarDAL, the tardal-dt pre-trained model given by the authors was used, and a timing code was added to its fusion script. The timing started at the beginning of the fusion process and ended before saving the image (excluding image saving time), with the timing setup for other models being similar. DenseFuse, by default, crops images to a resolution of 256×256, reducing runtime; we turn off image cropping in this study. MetaFusion defaults to cropping images to a resolution of 512×384; we modify it to keep the input and output resolution consistent for comparison with other algorithms. Considering that some models’ code frameworks are outdated and could not utilize GPU acceleration on our machine, all our models were run on the same computer with an Intel Core i7-9750H CPU @ 2.60 GHz processor, and GPU acceleration was turned off to ensure a consistent hardware platform for model execution.

It can be observed that our model operates quickly, processing a pair of images on average every second, but is slightly slower than methods like DenseFuse, SDNet, and PMGI in terms of speed. An interesting point to note is that the aforementioned three models are based on the TensorFlow code framework, whereas our model is built on PyTorch. Therefore, the slower performance of our model might be related to the internal implementations of the deep learning framework, although this requires further research. Although our model has the smallest number of parameters, the computational method of the transformer and the model’s recursive calling may also limit further speed improvements.

## 5. Conclusions

This paper proposes a lightweight and training-efficient infrared and visible image fusion model. The model only requires about 15 min of training on an NVIDIA GeForce RTX 3090 GPU, without the need for the demanding hardware specifications and potentially extensive tuning time costs associated with other larger models. Its end-to-end training approach is straightforward and effective, avoiding the complexity of two-stage training processes like CDDFuse and the instability issues associated with GAN-based models. Current researchers are keen on designing broader and deeper models, for instance, by incorporating self-attention mechanisms of transformer and dense connections to improve fusion effects. However, the performance improvement might be unacceptable compared to the added expense. As demonstrated in this paper, a few simple max pooling and average pooling layers can achieve satisfactory fusion results. Despite its minimal parameter count, the model’s processing speed has yet to reach its optimum. We consider further exploring the relationship between model parameters and processing speed in the future.

## Figures and Tables

**Figure 1 sensors-24-02466-f001:**
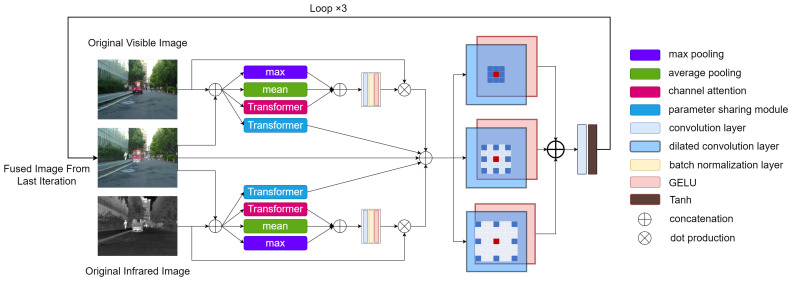
Overall flowchart of the model.

**Figure 2 sensors-24-02466-f002:**
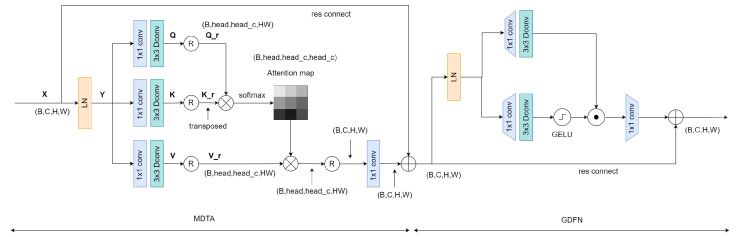
Structural diagram of the parameter sharing module and the transformer module.

**Figure 3 sensors-24-02466-f003:**
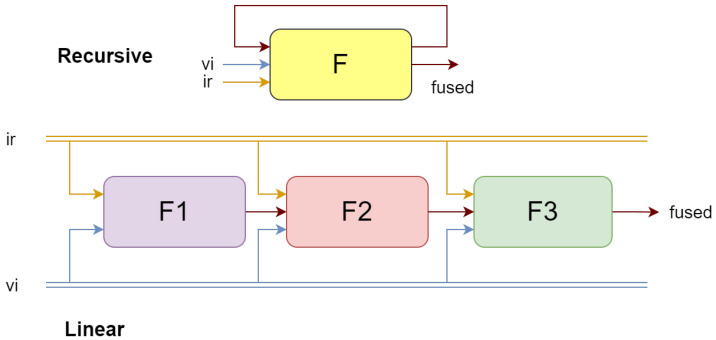
Schematic diagram of cyclic architecture and linear architecture.

**Figure 4 sensors-24-02466-f004:**
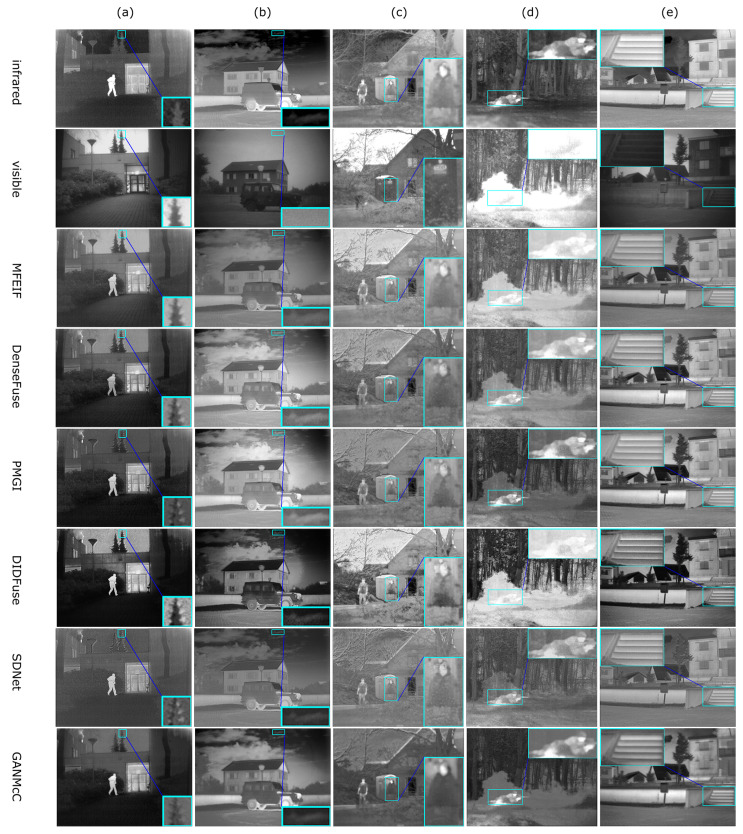
Comparison of visual effects with 9 models on the TNO dataset. (**a**) Kaptein 1123; (**b**) Marne 04; (**c**) two men in front of house; (**d**) soldier behind smoke; (**e**) Marne 07.

**Figure 5 sensors-24-02466-f005:**
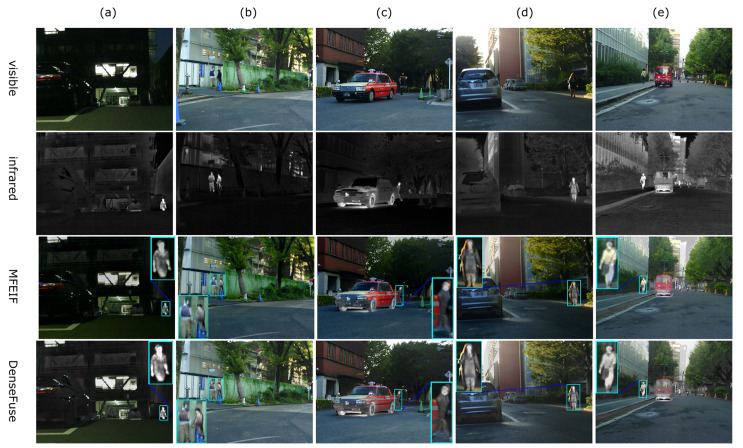
Comparison of visual effects with 9 models on the MSRS dataset. (**a**) 00051N; (**b**) 00085D; (**c**) 00131D; (**d**) 00169D; (**e**) 00357D.

**Figure 6 sensors-24-02466-f006:**
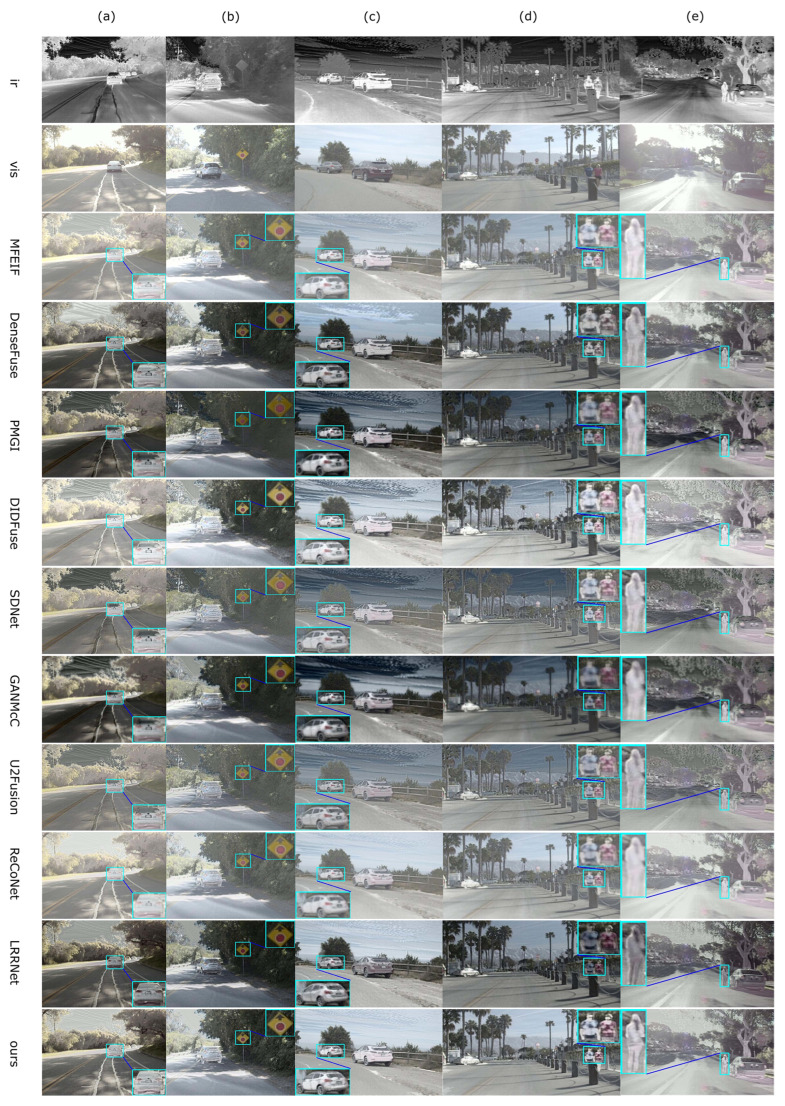
Comparison of visual effects with 9 models on the RoadScene dataset. (**a**) FLIR 00306; (**b**) FLIR 00497; (**c**) FLIR 01463; (**d**) FLIR 04269; (**e**) FLIR 04302.

**Figure 7 sensors-24-02466-f007:**
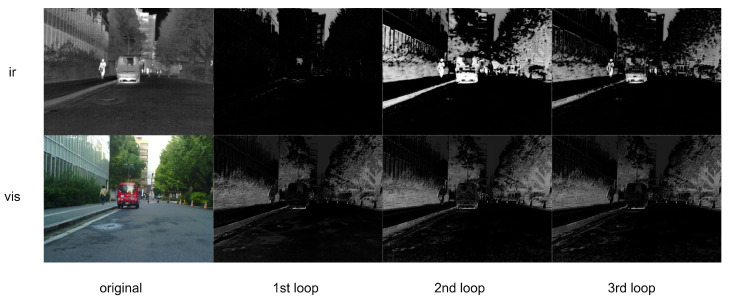
Visual result of our recurrent learning mechanism.

**Figure 8 sensors-24-02466-f008:**
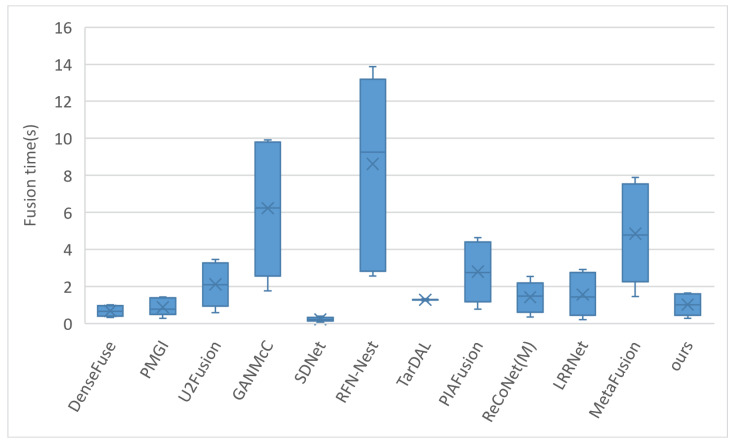
Comparison of fusion time for 20 pairs of images on the TNO dataset. The fusion time represents the average time taken to fuse each pair of images.

**Table 1 sensors-24-02466-t001:** Parameters of current advanced fusion models. The model we propose has the smallest number of parameters.

Model	Parameters *	Model	Parameters	Model	Parameters
DenseFuse	74,193	GANMcC	1,864,129	ReCoNet	7527
MFEIF	371,140	SDNet	67,091	LRRNet	196,816
PMGI	42,017	RFN-Nest	19,165,952	CDDFuse	1,188,272
U2Fusion	659,217	TarDAL	296,577	MetaFusion	811,714
DIDFuse	260,935	PIAFusion	1,266,595	Ours	5885

* All values in the ‘Parameters’ column represent the number of trainable parameters in each model.

**Table 2 sensors-24-02466-t002:** Comparison with 9 SOTA fusion methods on the TNO dataset; 20 pairs of images are selected for testing, and the results are averaged and rounded to two decimal places. ↑ indicates that the larger the indicator value, the better the fusion effect. Bold and $\underline{underline}$, respectively, indicate the best and second-best results.

	EN ↑	SD ↑	SF ↑	MI ↑	SCD ↑	VIF ↑	Qabf ↑
DenseFuse	6.86	35.59	9.33	1.54	$\underline{1.81}$	0.62	0.42
MFEIF	6.72	33.16	7.54	$\underline{1.73}$	1.74	$\underline{0.64}$	$\underline{0.45}$
PMGI	6.85	32.97	9.00	1.43	1.72	0.57	0.37
DIDFuse	$\underline{7.09}$	47.47	$\underline{11.78}$	1.62	1.83	0.60	0.41
GANMcC	6.72	32.85	6.44	1.54	1.71	0.50	0.26
U2Fusion	6.15	23.95	8.25	1.14	1.51	0.50	0.39
SDNet	6.29	23.98	9.86	1.28	1.46	0.48	0.41
ReCoNet	6.89	40.24	8.07	1.66	1.70	0.54	0.38
LRRNet	7.07	42.13	10.18	1.72	1.63	0.57	0.37
ours	7.13	$\underline{44.24}$	12.07	2.00	1.70	0.71	0.47

**Table 3 sensors-24-02466-t003:** Comparison with 9 SOTA fusion methods on the MSRS dataset; 20 pairs of images are selected for testing, and the results are averaged and rounded to two decimal places. ↑ indicates that the larger the indicator value, the better the fusion effect. Bold and $\underline{underline}$, respectively, indicate the best and second-best results.

	EN ↑	SD ↑	SF ↑	MI ↑	SCD ↑	VIF ↑	Qabf ↑
DenseFuse	$\underline{6.46}$	40.48	10.29	2.68	1.50	0.87	0.63
MFEIF	5.92	35.34	8.67	2.04	$\underline{1.58}$	0.72	0.54
PMGI	5.85	19.62	8.73	1.33	0.84	0.59	0.36
DIDFuse	4.54	33.68	$\underline{10.95}$	1.40	1.15	0.30	0.22
GANMcC	6.16	28.75	6.06	1.73	1.42	0.64	0.32
U2Fusion	4.80	21.30	7.60	1.28	1.04	0.45	0.34
SDNet	5.02	16.96	8.32	1.15	0.89	0.44	0.32
ReCoNet	5.27	43.20	10.72	1.77	1.41	0.51	0.42
LRRNet	6.19	32.14	8.83	2.11	0.89	0.55	0.46
ours	6.50	$\underline{41.93}$	11.73	$\underline{2.37}$	1.74	$\underline{0.86}$	$\underline{0.61}$

**Table 4 sensors-24-02466-t004:** Comparison with 9 SOTA fusion methods on the RoadScene dataset; 40 pairs of images are selected for testing, and the results are averaged and rounded to two decimal places. ↑ indicates that the larger the indicator value, the better the fusion effect. Bold and $\underline{underline}$, respectively, indicate the best and second-best results.

	EN ↑	SD ↑	SF ↑	MI ↑	SCD ↑	VIF ↑	Qabf ↑
DenseFuse	$\underline{7.26}$	$\underline{48.57}$	12.68	$\underline{2.40}$	1.51	$\underline{0.62}$	0.48
MFEIF	6.84	35.74	8.84	2.30	1.66	0.61	0.44
PMGI	7.20	45.04	10.54	2.24	1.68	0.56	0.40
DIDFuse	7.18	46.57	$\underline{14.01}$	2.07	1.76	0.58	0.44
GANMcC	7.30	47.96	9.41	1.97	$\underline{1.74}$	0.52	0.34
U2Fusion	6.58	30.82	11.40	1.91	1.42	0.52	0.45
SDNet	7.04	39.10	12.82	2.38	1.40	0.55	$\underline{0.46}$
ReCoNet	6.82	37.90	8.61	2.33	1.56	0.54	0.37
LRRNet	7.11	43.98	12.85	2.16	1.65	0.53	0.38
ours	7.25	48.85	14.75	2.50	1.62	0.67	0.44

**Table 5 sensors-24-02466-t005:** Comparison of two architectures when training on the MSRS dataset, with a batch size of 16, training for 30 epochs.

	GPU Memory (MB)	Model Parameters	Average Training Time per Batch (s)
Linear	61.66	17,655	47.0
Recursive	61.27	5885	45.8

**Table 6 sensors-24-02466-t006:** Comparison of two architectures when testing on the MSRS dataset.

	GPU Memory (MB)	Average Time for Fusing Each Pair of Images (s)
Linear	17.635	0.050
Recursive	17.545	0.051

**Table 7 sensors-24-02466-t007:** Comparison of the fusion effects of two architectures on the MSRS dataset.

	EN	SD	SF	MI	SCD	VIF	Qabf
Linear	6.34	38.89	9.67	2.16	1.59	0.75	0.55
Recursive	6.41	39.75	11.47	2.29	1.67	0.81	0.60

**Table 8 sensors-24-02466-t008:** Ablation experiment results on the MSRS dataset. Bold indicates the best result.

	EN	SD	SF	MI	SCD	VIF	Qabf
ori	6.36	39.87	10.12	2.00	1.71	0.77	0.57
+base	6.45	42.01	11.51	2.34	1.72	0.86	0.62
+trans	6.49	42.73	**11.96**	2.40	**1.76**	0.87	0.62
ours	6.61	43.52	11.80	2.99	1.66	0.92	0.63

## Data Availability

Publicly available datasets were analyzed in this study. The TNO dataset can be found at https://figshare.com/articles/dataset/TNO_Image_Fusion_Dataset/1008029 (accessed on 27 November 2023), the MSRS dataset at https://github.com/Linfeng-Tang/MSRS (accessed on 26 November 2023) and the RoadScene dataset at https://github.com/hanna-xu/RoadScene (accessed on 27 November 2023).

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
