# Peer review of "FERFusion: A Fast and Efficient Recursive Neural Network for Infrared and Visible Image Fusion"

_sensors, 2024, doi:10.3390/s24082466_

Round 1

Reviewer 1 Report

Comments and Suggestions for Authors

Thank you for submitting meaningful research results.

Research design, experiments, and analysis of results were carried out appropriately.

However, the conclusion is somewhat weak. It needs to be written by supplementing the key points.

Please review whether there is any overlap or unnecessary content overall, and emphasize the key points of the paper.

Thank you for your efforts.

Author Response

Thank you for your valuable feedback! We have found that the original discussion section indeed overlaps significantly with the abstract, so we have revised the discussion section as follows:

‘This paper proposes a lightweight and training-efficient model for infrared and visible light image fusion. The model only requires about 15 minutes of training on an NVIDIA GeForce RTX 3090 GPU, without the need for the demanding hardware specifications and potentially extensive tuning time costs associated with other larger models. Its end-to-end training approach is straightforward and effective, avoiding the complexity of two-stage training processes like CDDFuse, as well as the instability issues associated with GAN-based models. Current researchers are keen on designing broader and deeper models, for instance, by incorporating self attention mechanisms of Transformer and dense connections to improve fusion effects. However, the degree of performance improvement might be unacceptable compared to the added expense. As demonstrated in this paper, a few simple max pooling and average pooling layers can achieve satisfactory fusion results. Despite its minimal parameter count, the model's processing speed has not yet reached its optimum. We consider further exploring the relationship between model parameters and processing speed in the future.’

Reviewer 2 Report

Comments and Suggestions for Authors

The manuscript proposed a lightweight and effective image fusion model and tested the model performance on three benchmark datasets. Additional clarification should be provided on the following issues.

(1) What are the parameter units in table 1? And, the numerical format is not publication compliant.

(2) The elaboration about the motivation for the model development is insufficient, especially about the advantages of the model structure with respect to existing models.

(3) Line 263 ".... our model can directly take color images as input for training..." ... Why does Figure 2 show the using of 3 images, especially the use of color images of the two differences?

(4) The model flowchart does not characterize the model structure well, e.g. where is the Transformer module located?

(5) The model uses early-stage feature fusion, why isn't the type of operation performed in the middle or later stages? What is the evidence that early feature fusion is better?

(6) The presentation on the seven model performance evaluation metrics is too brief. Not all authors are familiar with the field of image fusion, and so the significance of the high and low values of the assessment metrics should be further explained.

(7) As far as the experimental results are concerned, the proposed model in the manuscript does not perform better than the DenseFuse model, especially in terms of operational efficiency. Why to claim that model is fast and efficient?

(8) Were only 3 rounds of recurrent learning performed? If not, deeper results should be shown. Also, the impact of the number of rounds on model performance can be better quantified using evaluation metrics rather than images.

(9) The discussion section is too brief and the manuscript lacks a conclusion section, which is not acceptable!

Reviewer 3 Report

Comments and Suggestions for Authors

This paper aims to address the challenges of extensive space requirements for storing network parameters and slow processing speeds encountered in existing infrared and visible image fusion methods. It proposes a lightweight, training-efficient recursive fusion neural network model for rapid and efficient infrared and visible image fusion. The topic is of great significance, the methodology is thoroughly described, and the results validate the effectiveness of the proposed method. However, there are some issues that should be clarified.

1.     Some descriptions in the paper lack appropriate references, e.g. "In fact, many attention mechanisms, due to their excessive focus on modally significant targets, tend to cause the loss of other information in the source images" in lines 128-130; "with the introduction of channel attention, enables the network to more flexibly perceive the relationships between different channels" in lines 152-153; it is better to provide references to support these claims. Including references for these claims would not only validate the assertions but also enhance the academic rigor of the paper.

2.     As mentioned in Section 3, attention mechanisms tend to cause loss of information in the source images. Recursive learning was used to address this issue. Could you provide some results demonstrating the effectiveness of this method in reducing information loss?

3.     The visual result of LRRNet seems to be better than your method, e.g. for the TNO dataset, please explain.

4.     There are three “Figure 3” in the paper.

5.     Why does the result shown in Figure 5 seem different from that shown in other figures, such as "Figure 3"?

Author Response

Dear professors:

     Thank you for taking the time to read our paper and for providing valuable feedback. Your suggestions have been immensely helpful in improving the quality of our paper. Below, we have responded to each of your comments and have made corresponding revisions. These changes have been highlighted at the end of the file for your review. Please see the attachment.

Reviewer 4 Report

Comments and Suggestions for Authors

This paper proposed FERFusion: A Fast and Efficient Recursive Neural Network For Infrared and Visible Image Fusion, the experimental data is convincing, but I still have some issues that the author needs to address:

(1) The author has defined multiple formulas as a single number in the formula numbering definition, such as equations (3), (4) etc. Please carefully check if it is reasonable.

(2) In section 4.1, the author converted RGB to YCrCb color space, why not YUV or HSI conversion, and what are the advantages of this conversion.

(3) In the Introduction section, the author needs to elaborate and supplement the application of traditional algorithms (such as rolling guidance filtering, shearlet, etc) in image fusion, such as the following literature, which the author needs to discuss and cite:

[1] Qi B, Bai X, Wu W, Zhang Y, Lv H, Li G. A Novel Saliency-Based Decomposition Strategy for Infrared and Visible Image Fusion. Remote Sensing. 2023; 15(10):2624.

[2] Liu Y, Wu Z, Han X, Sun Q, Zhao J, Liu J. Infrared and Visible Image Fusion Based on Visual Saliency Map and Image Contrast Enhancement. Sensors. 2022; 22(17):6390.

[3] Li L, Lv M, Jia Z, Ma H. Sparse Representation-Based Multi-Focus Image Fusion Method via Local Energy in Shearlet Domain. Sensors. 2023; 23(6):2888.

(4) The following relevant deep learning about Transformer algorithms need to be further discussed and cited in the article:

[1] Gao, Y.; Zhang, M.; Wang, J. Cross-Scale Mixing Attention for Multisource Remote Sensing Data Fusion and Classification. IEEE Transactions on Geoscience and Remote Sensing, 2023, 61, 5507815.

[2]  Zhang, X.; Li, Q. FD-Net: Feature Distillation Network for Oral Squamous Cell Carcinoma Lymph Node Segmentation in Hyperspectral Imagery. IEEE Journal of Biomedical and Health Informatics, 2024, 28 (3), 1552-1563.

(5) The year in reference 62 is incorrect, it should be 2022. The author should verify it, including accurate information from other references.

(6) The authors should be ranked according to the year of publication in Tables 1-4. Please update them.

(7) English grammar requires minor revision.

(8) The title of the reference section should be consistent in capitalization or lowercase, for example, references 1-4.

Comments on the Quality of English Language

 English grammar requires minor revision.

Author Response

(The authors gave the same response as above.)

Round 2

Reviewer 2 Report

Comments and Suggestions for Authors

(1) additional information on the motivation for the Transformer module followed by the Convolution module; and (2) the conclusion is an essential element of each paper.

Reviewer 3 Report

Comments and Suggestions for Authors

The authors have answered my concerns. 

Author Response

Many thanks!